# Uniparental Inheritance and Recombination as Strategies to Avoid Competition and Combat Muller’s Ratchet among Mitochondria in Natural Populations of the Fungus *Amanita phalloides*

**DOI:** 10.3390/jof9040476

**Published:** 2023-04-15

**Authors:** Yen-Wen Wang, Holly Elmore, Anne Pringle

**Affiliations:** 1Department of Botany, University of Wisconsin-Madison, Madison, WI 53706, USA; 2Rethink Priorities, San Francisco, CA 94117, USA; 3Department of Bacteriology, University of Wisconsin-Madison, Madison, WI 53706, USA

**Keywords:** population genetics, population genomics, mitochondrial recombination, mitochondrial inheritance, Muller’s ratchet

## Abstract

Uniparental inheritance of mitochondria enables organisms to avoid the costs of intracellular competition among potentially selfish organelles. By preventing recombination, uniparental inheritance may also render a mitochondrial lineage effectively asexual and expose mitochondria to the deleterious effects of Muller’s ratchet. Even among animals and plants, the evolutionary dynamics of mitochondria remain obscure, and less is known about mitochondrial inheritance among fungi. To understand mitochondrial inheritance and test for mitochondrial recombination in one species of filamentous fungus, we took a population genomics approach. We assembled and analyzed 88 mitochondrial genomes from natural populations of the invasive death cap *Amanita phalloides*, sampling from both California (an invaded range) and Europe (its native range). The mitochondrial genomes clustered into two distinct groups made up of 57 and 31 mushrooms, but both mitochondrial types are geographically widespread. Multiple lines of evidence, including negative correlations between linkage disequilibrium and distances between sites and coalescent analysis, suggest low rates of recombination among the mitochondria (ρ = 3.54 × 10^−4^). Recombination requires genetically distinct mitochondria to inhabit a cell, and recombination among *A. phalloides* mitochondria provides evidence for heteroplasmy as a feature of the death cap life cycle. However, no mushroom houses more than one mitochondrial genome, suggesting that heteroplasmy is rare or transient. Uniparental inheritance emerges as the primary mode of mitochondrial inheritance, even as recombination appears as a strategy to alleviate Muller’s ratchet.

## 1. Introduction

Mitochondria drive cellular respiration in nearly all eukaryotes [1]. Derived from an ancient endosymbiosis between an α-proteobacterium and the common ancestor of eukaryotes [1], mitochondrial genomes are inherited independently from nuclear genomes and, as a result, evolve on different trajectories [2].

Mitochondrial genomes typically mutate faster than nuclear genomes [3,4]. Higher mutation rates are likely the result of DNA damage caused by oxidation reduction, insufficient DNA repair mechanisms, and low replication fidelity, but other causes may contribute to the dynamic as well [5]. While estimates of mitochondrial mutation rates are generally higher than estimates of nuclear mutation rates [6], differences between mitochondrial and nuclear nucleotide diversities vary greatly among organisms [6].

Mitochondria are typically inherited from a single parent, preventing recombination [2]. Combined with background selection on linked sites, the lack of recombination can reduce effective population sizes, resulting in inefficacious selection (in contrast to low selection strength/coefficient) [7]. As a result, deleterious mutations can accumulate in mitochondrial genomes, causing an increase in genetic loads and the erosion of genomes through a process known as Muller’s ratchet [8,9]. Because of the generally elevated mutation rates in mitochondria, Muller’s ratchet may be especially relevant to the evolution of mitochondrial genomes. 

One solution to Muller’s ratchet may be mitochondrial recombination. To effectively recombine, two genetically distinct mitochondrial copies must first be found in the same cell, typically because distinct genome copies are biparentally inherited by an offspring (which would be heteroplasmic as a result). Biparental inheritance is rarely observed in plants and animals [10,11], but it can be found (e.g., in bladder campions and humans) and is well-known in the fungus *Saccharomyces cerevisiae* [12]. In this yeast, a rapid segregation of genetically distinct mitochondria is caused by the mitoses following plasmogamy [12], and segregation quickly renders the yeast cells homoplasmic. However, biparental inheritance and mitochondrial recombination may also impose costs. At evolutionary equilibrium, the fitness among individuals undergoing biparental inheritance may be more homogeneous than the fitness among individuals experiencing uniparental inheritance. The greater homogeneity of fitness among individuals can attenuate the efficiency of selection and emerges as a different mechanism leading to a greater accumulation of mutations [13,14,15]. Moreover, the proliferation of selfish mitochondria enabled by biparental inheritance and subsequent intracellular competition can also be harmful [16]. 

Mitochondrial inheritance among filamentous basidiomycetes typically follows one of three patterns. Many filamentous basidiomycetes appear to inherit their mitochondria uniparentally, even though nuclei migrate bilaterally following mating [17]: in the typical life cycle, two haploid mycelia mate by fusing bodies. Mated mycelia exchange nuclei reciprocally, but mitochondria are not exchanged [18,19,20]. The resulting dikaryotic mycelium thus contains the same set of two nuclear genomes in each cell, but different parts of the mycelium house different mitochondrial genomes [17]. However, unilateral nuclear migration associated with uniparental mitochondrial inheritance is also documented [17,21]. In these cases, only one of the haploid mycelia donates its nuclei, and the cells in the resulting dikaryotic mycelium share the mitochondrial genotype of the second haploid mycelium, which received the nuclei [20,21]. A third alternative is biparental inheritance: mitochondria are exchanged during mating, resulting in heteroplasmy. The exact dynamics of nuclear and mitochondrial exchange leading to heteroplasmy are unknown [22,23]. In some species, heteroplasmy is maintained throughout the life cycle, and even mushrooms possess two kinds of mitochondria (e.g., *Thelephora ganbajun*) [24]. Biparental inheritance provides an opportunity for recombination between mitochondria. In fact, because mitochondrial recombination is observed in the laboratory [22,23,25] and in natural populations [26,27,28], heteroplasmy may be a widespread (if potentially rare or transient) feature of mitochondrial dynamics in filamentous fungi. However, studies tracking the inheritance of mitochondria in natural populations are rare, and we do not have a strong understanding of mitochondrial evolution in nature.

Mycologists have not compared the mutation rates of nuclei and mitochondria with mutation accumulation experiments, nor used crossing studies or phylogenetic approaches. The limited data available suggest that the genetic diversity of mitochondrial DNA is lower, as compared with the genetic diversity of nuclear DNA [27,29,30]. In addition to the mutation loads caused by single-nucleotide mutations and indels, homing endonucleases in introns are often found in fungal mitochondria and may also have a deleterious effect on an organism [31]. Homing endonucleases act similarly to transposases [32,33], cleaving DNA at “homing sites” and subsequently inserting into other “homing sites” through homologous recombination or retrotranscription [32]. Homing endonucleases are often considered to be neutral or even beneficial to the host genome because they self-splice [34], but off-target recognition and insertion may cause damage and has been observed in mutagenesis experiments in bacteria [35].

The filamentous basidiomycete fungus *Amanita phalloides* (the death cap) is native to Europe and invasive in California [36]. It produces a broad array of toxins and often causes human and animal deaths [37]. The mitochondrial genomes of *Amanita* species are characterized by architectural rearrangements, including a recent translocation in *A. phalloides* [38]. In addition, intron patterns are diverse among the *Amanita*, with *A. phalloides* having the highest intronic content [38]. Recently, we generated a population genomics dataset of 88 sequenced *A. phalloides* mushrooms; the 88 mushrooms resolve into 39 genetic individuals [39]. Our collection also includes two unisexual individuals, each of which was generated from a single (haploid) parental mycelium. The dataset presents a unique opportunity to study the evolution of mitochondrial genomes in a natural population.

To elucidate how the mitochondria of this filamentous basidiomycete resist Muller’s ratchet in nature, we systematically investigated patterns of genetic diversity and recombination, and potential modes of inheritance. Using the population genomics dataset, we aimed to (1) understand the mitochondrial genome architecture of *A. phalloides*, (2) test whether individuals are mitochondrial mosaics, (3) compare the genetic diversity and signals of selection between the mitochondrial and nuclear genomes, (4) test for recombination and estimate the recombination rate of the mitochondrial genome, and (5) identify whether there is evidence for biparental inheritance. The population dynamics of *A. phalloides* mitochondria offer a holistic view of how one species’ mitochondrial genomes combat Muller’s ratchet.

## 2. Materials and Methods

### 2.1. Collecting Mushrooms, and Sequencing of Genomes and a Transcriptome

Our collecting strategy, and genome and transcriptome sequencing protocols, are described in Wang et al., 2023 [39]. In brief: We collected multiple specimens (mushrooms) of *A. phalloides* from two sites in California (an invasive range; Drake2 and Drake3; N = 67), and three sites in Portugal (a native range; Agraria, Mira, and Vilarinho; N = 11). We collected in Portugal in 2015, but the California samples were collected in 2004, 2014, and 2015 (N = 19, 34, and 14, respectively; Appendix A). We also sequenced herbarium samples from across Europe (N = 9) and another older herbarium sample from California (N = 1). Some of the mushrooms derive from the same genetic individuals (a single mycelium) and contain the same genome (a total of N = 88 mushrooms = 39 genetic individuals; including 29 single-mushroom individuals and 10 two–nineteen-mushroom individuals [39]). Two samples from Spain and California were used in this study but were not included in Wang et al., 2023 because the sequence data were of insufficient quality for the nuclear variant calling used in Wang et al., 2023 (samples 10003 and 10220). No other mushrooms were collected from the same region as sample 10003 and it is its own individual. AFLP data suggested that 10220 also belongs to its own individual (Golan, personal communication). We sequenced all samples with the Illumina Hiseq2500. We chose Portuguese sample 10511 to serve as a reference genome due to its high DNA yield, and we also sequenced this sample with Pacbio RS II Sequel to enhance scaffolding. Nuclear genome assembly and variant calling are described in Wang et al., 2023. To annotate the genome, we sequenced the transcriptome of California sample 10721 using an Illumina HiSeq2500 platform using a 350 bp insert and 126 bp paired-end reads, and we trimmed the raw reads with Trimmomatic ver. 0.35 [39]. 

### 2.2. Reference Mitochondrial Genome Assembly and Annotation 

To elucidate mitochondrial genome architecture, we de novo assembled the mitochondrial genome of our reference specimen (10511) using NOVOPlasty ver. 2.7.2 [40]. We used a k-mer size of 39 and a previously published *A. phalloides* mitochondrial large subunit ribosomal RNA partial sequence (Acc: HQ540033.1) as the seed to initiate the assembly. We error-checked our assembly by re-mapping Illumina raw reads to the assembly using the MEM algorithm in BWA ver. 0.7.17 [41]. The final assembly is deposited in NCBI GenBank under accession number MW436401.

To annotate the mitochondrial genome, we used the online version of MFannot [42] (last accessed: 5 July 2019). We translated predicted genes into protein sequences using EMBOSS ver. 6.6.0.0 using a mold/protozoan mitochondrial DNA codon table [43]. We functionally annotated these protein-coding genes using Interproscan 5 [44] and BLASTp ver. 2.9.0+ [45] (last accessed: 17 April 2020). We refined our annotations of tRNAs using tRNAscan-SE ver. 2.0.5 [46] (last accessed: 6 March 2020).

Because MFannot does not annotate rRNA genes, we assembled the de novo transcriptome from sample 10721 with Trinity ver. 2.2.0 [47]. We then searched for rRNA assemblies within the assembled transcriptome using BLASTn, using partial nucleotide sequences of the *A. phalloides* mitochondrial large subunit and mitochondrial small subunit rRNA (Acc: HQ540033.1 and HQ539933.1, respectively) as queries. To identify introns, exons, and gene boundaries of the mtLSU and mtSSU genes, we used BLASTn to align the best hits to our reference mitochondrial genome assembly. 

We then explored the gene expression levels. To improve transcriptomic raw read mapping, we first assembled the mitochondrial genome of sample 10721 following the same strategy described above. To ensure an accurate alignment between the assemblies of 10721 and our reference specimen, we rotated the assembly of 10721 by identifying loci matching the start of our reference specimen’s assembly. 

We used MFAnnot to annotate the assembly of 10721, and we used BLASTp to match the annotations to the annotations of the reference specimen. Finally, we mapped the transcriptomic raw reads to the 10721 assembly with HISAT2 ver. 2.1.0 [48] (maximum intron length = 10,000 bp). After removing unmapped and duplicated raw reads as needed, we calculated the FPKM (fragment per kilobase of transcript per million mapped reads) for each gene with Cufflinks ver. 2.2.1 [49].

### 2.3. Assembling the Mitochondrial Genomes of the Remaining Samples, Haplotype Network Reconstruction, and Nuclear Phylogeny Reconstruction

We assembled and rotated the remaining 86 mitochondrial genomes using the same strategy described for sample 10721. Most genomes assembled into a single contig. For multicontig assemblies, we scaffolded the assemblies by concatenating contigs according to the configuration suggested by NOVOPlasty. We then aligned the different assemblies with MAFFT ver. 7.455 [50]. To understand whether the multiple mushrooms of any genetic individual house multiple types of mitochondria, we identified invariable haplotypes (mitotypes) for each of the 88 mushrooms based on the mitochondrial alignment among all samples. If a genetic individual appeared to have multiple mitotypes, then we examined the differences manually; we considered only those SNPs not associated with single-nucleotide repeats or problematic gappy regions. If our manual inspection provided no confidence in the presence of multiple mitotypes, then the most prevalent mitotype was used for each genetic individual in subsequent analyses. To visualize genetic relationships among individuals, we reconstructed a clone-corrected mitochondrial haplotype network with the R package pegas [51]. 

To compare the mitochondrial haplotype network with relationships among nuclei, we calculated the Euclidean distances among individuals using the filtered, clone-corrected nuclear VCF described in Wang et al., 2023 [39]. An unrooted neighbor-joining tree was then reconstructed with the R package ape [52]. 

### 2.4. Comparing Mitochondrial and Nuclear Genomes

Our initial collecting focused disproportionately on geographically proximate invasive populations in California, and in subsequent analyses we focused on a subset of the data; the subset includes one California sample, two Portuguese samples from different populations (separated by >100 km), and the nine samples collected from throughout the rest of Europe (N = 12; Appendix A). 

To investigate the mitochondrial mutation rate, we compared the mitochondrial and nuclear nucleotide diversity (π). We first used the R package *pegas* to estimate the overall mitochondrial π (excluding sample 10003 because it was not included in Wang et al., 2023 [39]). To estimate the nuclear π, we filtered any variable site with a single occurrence of the minor allele, as well as sites overlapping with transposable elements, as detected by REPET ver. 2.5 [53]. We then retrieved the individuals of our subset from the larger dataset of 88 specimens and calculated the π for every 1000 bp block of nuclear alignments using VCFtools [54]. The average of the π was corrected for the transposable element masks, but not for the removal of single minor alleles (which will result in a slight underestimation of the π).

Because genic and nongenic regions experience different selective pressures (which can influence the patterns of π), we next focused on the nonsynonymous (π_n_) and synonymous (π_s_) diversity of the mitochondrial and nuclear subunits of three respiratory complexes (I, IV, and V). Because the subunits of each complex are encoded in both mitochondrial and nuclear genomes, and because the subunits of each complex coordinate to form a single functional protein, we assume the proteins within each complex experience similar selective pressures (Appendix A). Estimating π_n_ and π_s_ requires DNA alignments. To generate DNA alignments for mitochondrial genes, we used MAFFT to align protein sequences and reverse-translated the alignments to DNA alignments. To generate DNA alignments for nuclear genes, we first phased the nuclear VCF file from Wang et al., 2023 [39] with WhatsHap ver. 0.17 [55], and we then removed any variable sites with a single minor allele. WhatsHap did not fully resolve the haplotype phasing among variable sites, and so we first identified the loci of genes within the three respiratory complexes, and then retrieved the variants (including SNPs and indels) from these loci. Next, we fully phased the VCF file by randomly assigning phases between pairs of unphased variable sites, and then we retrieved and realigned each gene with bcftools and MAFFT. After retrieving the DNA alignments of both mitochondrial and nuclear genes, we used MEGA-CC ver. 10.1.5 [56] to calculate the π_n_ and π_s_ using a modified Nei–Gojobori method with a transition/transversion ratio of two. To estimate average levels of the π_n_ and π_s_ from all genes of the respiratory complexes, and then within each protein complex, we reran analyses on concatenated alignments. To statistically test for differences between the π_n_/π_s_ ratios of mitochondrial and nuclear genes, we bootstrap-sampled the concatenated mitochondrial and nuclear alignments of each of the three complexes 100 times and estimated the π_n_/π_s_.

### 2.5. Testing for Recombination among Mitochondrial Genomes

To test for recombination (assuming an infinite-sites model), we first explored every pair of biallelic sites among genetic individuals. We interpreted the presence of four different haplotypes for a pair of biallelic sites as evidence of recombination. We defined a haplotype block as a consecutive fragment of the mitochondrial genome for which fewer than four haplotypes are present, and for which any of the two biallelic single-nucleotide positions flanking the fragment can introduce a fourth haplotype (c.f., Wang et al., 2002 [57]). 

We then formally tested for recombination using the mitochondrial alignment of the data subset with the addition of sample 10003, and three strategies: first, we tested for correlations between linkage disequilibrium and the physical distance between segregating sites using Mantel tests; second, we tested whether, within a window, the ratio of segregating sites of the first few sites and that of the rest of the sites are contingent using maximum χ^2^ tests [58,59]; third, we tested whether the minimum number of recurrent mutations deviates from the expected value using PHI tests [60]. 

To calculate the physical distance between segregating sites, we used RecombiTEST [61] to run LDr^2^ and LDD’. Because Mantel tests require linear (noncircular) sequences, the tests were conducted on ten 10 kbp and five 20 kbp chunks of the whole mitochondrial alignment, resulting in a total of 30 tests on the same data subset used in the former section but including sample 10003. To avoid detecting any activities tied to the homing endonucleases of the mitochondrial genome [38], we performed maximum χ^2^ tests on both the whole mitochondrial genome and a homing-endonuclease-free region (from 70 to 95 kbp). We set the window size to 2/3 the number of segregating sites [62]. Finally, we performed PHI tests on the same homing-endonuclease-free region using SplitsTree ver. 4.14.8 [63]. 

To identify recombination breakpoints, we used a phylogenetic algorithm to analyze the alignment of the mitochondrial genomes. We identified the best nucleotide substitution model with ModelTest-NG [64] based on AICc, and we used the model to search for recombination breakpoints using GARD [65] implemented in HYPHY ver. 2.5.1 [66].

To estimate recombination rates, we used LDHAT ver. 2.2 [67]. First, we used the function COMPLETE to generate a likelihood lookup table by simulating populations with a population mutation parameter, (θ) = 0.003 (based on our previously estimated π), and population recombination parameter, (ρ) = 0–100. We then searched for recombination rates using the function INTERVAL to sample every 5000 iterations for one million iterations of a Markov chain Monte Carlo with a block penalty of 5. We discarded all samples from the burn-in period, equivalent to the first 250,000 iterations. We estimated the recombination rate to mutation rate (r/m) by calculating the mean of ρ/θ. To explore the recombination rate variation across the mitochondrial genome, we also calculated the ρ every 2000 bp (using the reference genome).

### 2.6. Testing for Biparental Inheritance of Mitochondria

To test whether there is evidence for biparental mitochondrial inheritance, we modified methods from Pyle et al., 2015 [68] to identify potentially different sources of the raw reads used to generate the mitochondrial genomes of individual mushrooms. Because we sampled intensively in California at site Drake2, we focused on Drake2, hypothesizing that potential biparental mitochondrial inheritance at the site would be generated from observed mitotypes. In other words, if any Drake2 individual housed two mitotypes, then the raw reads of its mitochondria would likely match two of the known and assembled mitotypes. The mitotype with the highest coverage would be the ultimately assembled mitotype, and we would interpret it as the “major” type. The mitotype with lower coverage would be the source of unassembled raw reads and would be interpreted as the “minor” type.

To use this approach, we first built libraries of 251 bp k-mers (the length of the original raw reads) from the Drake2 mitochondrial assemblies. Given two mitotypes, *x* and *y*, we defined a set of k-mers, *K_xy_*, including all k-mers of type *x* having at least two different nucleotides from any 251 bp k-mers of type *y* (discarding k-mers spanning any alignment gaps). For each mushroom, we identified the major type (*M*) (the assembly type) and chose a random mitotype (*i*) as a putative minor type. We then defined the sample’s raw reads in the sets *K_iM_* and *K_Mi_* as *R_iM_* and *R_Mi_*, and estimated putative heteroplasmy for the minor type *i* (${\hat{H}}_{i}$) as follows:(1)$${\hat{H}}_{i}=\frac{{\#R}_{iM}/{\#K}_{iM}}{{\#R}_{iM}/{\#K}_{iM}+{\#R}_{Mi}/{\#K}_{Mi}}$$

We calculated the ${\hat{H}}_{i}$ using each potential minor mitotype (each mitotype present at Drake2). Because the true detectable heteroplasmy should produce a higher ${\#R}_{iM}/{\#K}_{iM}$ than those from sequencing errors, we estimated the heteroplasmy ($\hat{H}$) of a sample with the following:(2)$$\hat{H}=\left\{{\hat{H}}_{i}\right|{\#R}_{iM}/{\#K}_{iM}=max\{{\#R}_{1M}/{\#K}_{1M},\ldots ,{\#R}_{nM}/{\#K}_{nM}\}\}$$

As a comparison, we also analyzed the European genomes generated from herbarium specimens (in this analysis, excluding the Portuguese samples collected in 2015). Each of the European mushrooms was collected in a different country (Appendix A), and the probability of discovering heteroplasmy with our approach in these samples is lower compared with the probability of discovering heteroplasmy at Drake2. We inferred putatively heteroplasmic samples by first fitting heteroplasmy to a beta distribution using the R package *extremevalues*, and then looking for right-tailed outliers (rho = 1). We compared the median heteroplasmy from Drake2 with the median heteroplasmy from the European samples using a Mann–Whitney test.

## 3. Results

### 3.1. Reference Genome Architecture and Gene Expression

The assembly of the reference mitochondrial genome is 100,852 bp and consists of a fully circularized single contig (Figure 1). The coverage depth is 2128.1. Mapping the raw reads to the assembly revealed no mismatches, establishing the assembly as complete and the DNA sequence as accurate. The GC content of the assembly is 24.6% (with a standard deviation (based on sliding windows) of 3.67%). The genome encodes 49 predicted protein-coding genes, including 15 core genes, 15 homing endonuclease genes, 2 noncore genes, and 17 hypothetical genes. A total of 2 rRNA and 27 tRNA genes are also present in the genome (Appendix A). Four protein-coding genes (*COX1*, *COX2*, *COB*, and *NAD5*) and the two rRNA genes have from one to nine introns. Most of the introns contain one homing endonuclease, a common characteristic of fungal mitochondrial introns [31]. 

The LSU and SSU rRNA genes are the most expressed genes, with FPKM values of 120,284 and 90,088, respectively. Following the rRNA genes, the five most highly expressed genes are *COX4*, *ATP9*, *NAD3*, *ATP6*, and *ORF179* (FPKM = 5659, 2764, 2360, 2326, and 2114, respectively) (Figure 2). Core genes have higher FPKM values compared with noncore, homing endonuclease, and hypothetical genes (Dunn’s post hoc test *p*-value < 0.05), whereas noncore, homing endonuclease, and hypothetical genes are not significantly different from each other. Certain homing endonuclease genes also have high FPKM values (e.g., *ORF179* encodes a homing endonuclease), and these homing endonucleases are located in introns of rRNA genes. We hypothesize that the seemingly high expression levels of these homing endonuclease genes are caused by their presence within the introns of rRNA genes; they are likely expressed as pre-rRNA but remain within introns after the introns are cleaved. We note that our transcriptomic protocols were originally intended to generate sequences from nuclear mRNA and so we used a poly-A pulldown assay. Our expression estimates may reflect the protocol’s bias to nuclear mRNA, as mitochondrial RNAs generally lack poly-A tails. Nonetheless, if all mitochondrial transcripts lack poly-A tails, then the pulldown assay should inflict little bias among the expression levels of mitochondrial genes. 

### 3.2. Structural Diversity 

Each of the mitochondrial genome assemblies is circularized and the sequencing depths range from 241.5X to 2925.0X (Appendix A). All but three assemblies encompass a single contig, and the three atypical assemblies are made up of two contigs. The atypical assemblies stem from three mushrooms of a single individual collected in Vilarinho, Portugal. Contigs of these samples break at the poly-C and AT-rich regions; these regions are often problematic in genome assemblies. The poly-C region also introduces errors in assemblies. Because there is no other polymorphism outside of this region among these three assemblies, we concluded that these three assemblies represent the same mitochondrial genome.

Different mushrooms sampled from the same genetic individual typically have an identical assembly, or mitotype. However, four genetic individuals appear to present different mitotypes in different mushrooms. However, differences between mushrooms consist of small alignment gaps, likely the result of sequencing slippage. We conclude that individuals with apparently different mitotypes likely possess identical mitochondrial genomes.

Most of the mitochondrial genomes are similar in size. Genome sizes of all but seven mushrooms range from 100,799 to 100,921 bp, with a mean of 100,852.7 bp (Appendix A). The three smallest genomes (mushrooms 10003, 10004, and 10513; from 94,933 to 95,145 bp) each have one of three different deletions between 15,732 and 22,048 bp (deleted length = from 5734 to 5924 bp). The three deletions are all at the same site, overlapping with two hypothetical genes and a noncore gene (*POLB/ORF129*; encoding a DNA polymerase-type beta). The three largest assemblies (mushrooms 10506, 10508, and 10509; from 104,034 to 104,075 bp), belonging to a single individual, have an identical insertion at 34,538 bp (insert length = 3118 bp); these are also mushrooms from Vilarinho with assemblies made up of two contigs. The smaller contigs do not overlap with the large insertions but locate close to a repetitive region. In addition, the genome of mushroom 10018 is also rather large (102,773 bp). It has a complex polymorphism between 19,519 and 19,770 bp, likely the result of an insertion and deletion (insert length = 2177 bp).

To investigate the sources of insertions, we searched for homologous sequences of the inserts of mushrooms 10018 and 10508 by reannotating their mitochondrial genomes and BLASTing against the reference genome. In sample 10018, the insert contains an open reading frame (ORF) and a tRNA. The ORF is homologous to *ORF152*, encoding a hypothetical protein (similarity = 98.7%). The tRNA is likely to be a pseudogene (it has an undetermined anticodon) and its origin is unclear. In sample 10508, the insert disrupts *NAD6* and contains three ORFs: two homing endonucleases, homologous to *ORF166* and *ORF361*, and one hypothetical gene, homologous to *ORF162*. Interestingly, these three genes are at least 20 kbp apart from each other, and the two homing endonucleases are less similar to their homologous genes (similarity = 65.8% (*ORF166*) and 71.1% (*ORF361*)) than the hypothetical gene is to its homologous gene (similarity = 93.8% (*ORF162*)). The distance between the three ORFs and differences in similarity may suggest an insertion stemming from three insertion events.

We visualized the relationship among mitotypes by building a haplotype network of the mitochondrial genomes of each genetic individual (N = 39, Figure 3). Mitochondrial genome diversity exhibits little spatial structure on a global scale, which is a stark contrast to nuclear genomes (Appendix A). Two major groups of mitotypes are differentiated by the region between 79 and 82 kbp. From this point forward, we refer to the group including the reference mitotype (N = 26) as the “reference group”, and to the second group as the “diverged group” (N = 13). While the nuclear genomes of Portugal cluster together and apart from the Californian genomes, both mitochondrial groups are found in both Portugal and California. However, only one of the European herbarium samples, from Spain, belongs to the diverged mitochondrial group. To investigate the source of the divergent 79–82 kbp region, we used the sequences of the two groups as a query in a search of NCBI’s GenBank database. The best and only hit for the 79–82 kbp region of both groups is the mitochondrial genome of *Amanita phalloides*, which suggests that divergence is not the result of horizontal gene transfer and instead stems from a recombination event with a diverged (and as yet unsequenced) mitochondrion of *A. phalloides* or another closely related species.

### 3.3. Nucleotide Diversity and Efficacy of Selection in Mitochondrial and Nuclear Genomes

Mitochondrial π is 0.00298, and nuclear π is 0.00306. Because the 79–82 kbp region of the mitochondrial genome is highly variable (Figure 3) and unlikely to accurately reflect the mutation rate, we removed it and estimated a π of 0.00125. The parameter π can be used to estimate the mitochondrial population mutation parameter, θ, enabling us to estimate the product of N_e_ and µ [69]. Assuming that *A. phalloides* inherits mitochondria uniparentally and migrates nuclei bilaterally, the expected mitochondrial and nuclear θ are 2N_e_µ and 4N_e_µ, respectively [70]. The N_e_µ of mitochondria (0.00063) is thus slightly lower than the N_e_µ of nuclei (0.00077).

As mitochondrial and nuclear genomes may contain different proportions of coding, noncoding, and other (e.g., nongenic) regions, simply comparing the nucleotide diversity of entire genomes may be misleading. To avoid potentially confounding variables, we estimated the π_n_ and π_s_ of the respiratory complexes. Using the same rationale as we used for entire genomes, we can estimate the product of N_e_ and the synonymous mutation rate (N_e_µ_s_), and the product of N_e_ and the nonsynonymous mutation rate (N_e_µ_n_). Separating the estimates for synonymous and nonsynonymous mutation rates allows us to distinguish neutral mutations from mutations under the influence of selection. The mitochondrial N_e_µ_s_ is half of the nuclear N_e_µ_s_, whereas the mitochondrial N_e_µ_n_ is twice the nuclear N_e_µ_n_ (Appendix A). After removing the outlier gene *NAD6*, the estimated mitochondrial N_e_µ_s_ is a quarter of the nuclear N_e_µ_s_, and the mitochondrial N_e_µ_n_ is half of the nuclear N_e_µ_n_ (Figure 4). Whether we remove the outlier or not, the mitochondrial π_n_/π_s_ is approximately twice or more of the nuclear π_n_/π_s_, and a higher mitochondrial than nuclear π_n_/π_s_ is supported by 98 out of 100 bootstrap analyses. We hypothesize that the lower N_e_µ_s_ of mitochondrial genes is caused by a smaller effective population size, which, in turn, is caused by a lower recombination rate and stronger background selection. The smaller effective population size due to background selection also results in inefficacious selection on nonsynonymous sites.

### 3.4. Detecting Mitochondrial Recombination and Estimating Rates 

We first tested for mitochondrial recombination with four-gamete tests. A total of 17,095 of 303,810 biallelic site pairs (5.6%) consist of four different haplotypes, suggesting recombination under an infinite-sites-model assumption. There are 40 haplotype blocks ranging in size from 35 to 13,023 bp. 

The ability of *A. phalloides* mitochondria to recombine is also supported by multiple tests under a finite-sites model (Table 1). Although not all tests yielded significant results, we found mitochondrial recombination highly plausible: (1) more than 70% of the LDr^2^ tests detected significant evidence of recombination; (2) a total of 60% of the LDD’ tests detected significant evidence when mitochondrial alignments were cut into 20 kbp; (3) both maximum χ^2^ tests yielded *p*-values = 0. However, using 10 kbp chunks of mitochondrial alignments, the LDD’ tests only detected evidence of recombination in one out of ten windows. The failure to detect recombination was likely caused by a lack of genetic diversity in the 10 kbp chunks; moreover, D’ is strongly skewed when allele frequencies are low [71,72]. We also failed to detect significant evidence of recombination using the PHI test.

To pinpoint the sites most likely to be involved in recombination, we used the GARD algorithm. It identified eight recombination breakpoints in the reference genome (Figure 1). The first and second breakpoint clusters (at 19,496, 20,252, 20,917 bp and 34,542, 36,204 bp, respectively) are found in loci associated with the complex mutation of mushroom 10018; these two breakpoint clusters are unlikely to be accurate because the GARD algorithm does not take indels into account. The last breakpoint cluster (76,740, 79,657, 81,531 bp) coincides with the 79–82 kbp region characterizing the reference and diverged haplotype groups (Figure 3) and it may be an explanation for the region’s distinctively high nucleotide diversity. Lastly, the breakpoint at 62,833 bp is found in the intron of the SSU rRNA. 

Because the complex mutation in sample 10018 likely introduces artifacts to estimates of the recombination rate, we removed sites between 19519 and 19770 bp and estimated a mean ρ = 3.54 × 10^−4^ per nucleotide and an r/m = 0.015 (ρ/θ_s_) or 0.125 (ρ/π). The mean ρ of every 2000 bp chunk ranges from 1 × 10^−5^ to 5.94 × 10^−3^ per nucleotide (Figure 5). The largest mean ρ peak coincides with the complex mutation and the first breakpoint cluster identified by GARD. Interestingly, the 79–82 kbp region has a relatively low ρ, suggesting that the statistical power of this region for GARD derives from its sequence divergence but not a high recombination rate. Nevertheless, the low recombination rate may explain the pattern of nucleotide diversity we observe in the mitochondrial genomes of *A. phalloides*.

### 3.5. Testing for Biparental Inheritance of Mitochondria

Heteroplasmy estimates for mushrooms collected from Drake2 range between 0% and 1.337% (median = 0.051%), and for European collections, between 0.004% and 0.272% (median = 0.011%) (Figure 6, Appendix A). We can only detect mitotypes for which we have genomes. Because the mitotypes at Drake2 were intensively sampled, we hypothesized that the heteroplasmy estimates would be higher at Drake2 compared with the European collection. (European herbarium specimens were collected very far away from each other (>25 km), and the mitotypes of their populations were not thoroughly characterized.) However, the estimates for these two collections are not significantly different (Mann–Whitney *p*-value = 0.0748), suggesting that biparental mitochondrial inheritance is not common. We found an outlier from Drake2: sample 10299 (1.337%), but also an outlier in the European collection: sample 10169 (0.272%).

Mushroom 10299 was generated by the same genetic individual as generated five other mushrooms (a homokaryotic mycelium, as reported by Wang et al., 2023 [39]). However, these five mushrooms do not have the same high degree of heteroplasmy as mushroom 10299 (mean = 0.056%) (Figure 6, Appendix A), suggesting that either their mitochondria have undergone enough bottlenecks to eliminate heteroplasmy, or mushroom 10299 has a relatively high background contamination rate compared with most other samples. We favor the second hypothesis.

## 4. Discussion

We systematically investigated the structure and diversity of 88 newly assembled *A. phalloides* mitochondrial genomes. By exploring multiple aspects of mitochondrial population genomics, including potential for recombination and mode of inheritance, we provide a comprehensive analysis of the relationship between mitochondrial dynamics and Muller’s ratchet in natural populations of a filamentous basidiomycete. The evolution of death cap mitochondria is characterized by structural rearrangements, recombination, evolutionary trajectories distinct from the trajectories that characterize nuclear genomes, and uniparental inheritance.

### 4.1. Mitochondrial Genome Architecture and Structural Diversity 

Among species within the genus *Amanita*, the mitochondrial genomic organization is dynamic and characterized by multiple translocations, inversions, and gains and losses of introns [38]. The organization of our reference genome matches the mitochondrial genome described in Li et al., 2020 [38]: it houses 49 protein-coding, 2 rRNA, and 27 tRNA genes. The two genomes house the same number of introns, and the introns are found in exactly the same places. However, our other death cap mitochondrial genomes are characterized by many differences. Mitochondrial genomic organization within a species appears to be as dynamic as organization across species. 

We identified three mitochondrial assemblies with large insertions, three mitochondrial assemblies with large deletions, and one mitochondrial assembly with a complex structural mutation that is difficult to interpret. The three assemblies with large insertions belong to a single individual collected from a single site in Portugal and the variation locates at a poly-C region, indicating that they likely represent the same mitochondrial genome. This mitochondrial genome is significantly larger than other mitochondrial genomes. The insertion resides in gene *NAD6* as an intron. The homologs of the genes found within the insertion (two homing endonucleases and one hypothetical gene) locate at distant loci, and each gene possesses a different degree of divergence to its homolog. Based on these data, we infer multiple insertion events at the site that introduce extra introns to the gene. Similar dynamics are reported in other fungal species; for example, in *Cryptococcus neoformans* and *Tremella fuciformis*, the mitochondrial genome size is highly correlated with the number of introns [73,74]. Homing endonucleases spread across populations at a much faster speed than other kinds of mutations [75]. If the homing endonucleases we observe are still active, then temporal sampling at a single site may allow us to observe the dynamics of homing endonucleases in natural populations. However, we did not see any difference in the numbers of endonucleases among the California collections from 2004, 2014, and 2015, suggesting that either the genes are not active, or their spread takes longer than a single decade.

The deletions characterizing a different set of three genomes, collected in Spain, Sweden, and Portugal, and the complex structural mutation characterizing a genome collected in Italy, are not associated with homing endonuclease genes; instead, they span a region that, in the reference genome, is made up of three genes: two hypothetical genes and one *POLB* gene. Genome size variation resulting from mechanisms other than homing endonucleases have also been reported in *Phellinus noxius*, where size variation is also associated with an intergenic region, in addition to its numbers of introns [76]. Interestingly, the deletions and complex structural mutation overlap with *POLB*. Although *POLB* is not considered as a core gene, the encoded DNA polymerase is suggested to play an important role in single-nucleotide base excision repair [77]. We hypothesized that the gene of these individuals was transferred into the nuclear genome, but we did not find it in the nuclear genome of any of the mushrooms. For the moment, we do not know whether the observed structural rearrangements impact the fitness of individuals.

### 4.2. Rare Recombination Contributes Lower Nucleotide Diversity and Selection Efficacy in Mitochondria as Compared with Nuclei

Mitochondria are often thought to be free of recombination, resulting in Muller’s ratchet—the irreversible accumulation of slightly deleterious mutations. However, recombination among fungal mitochondria has been reported in studies using multiple approaches, including experimental mating between strains with different mitochondrial markers [22,23,25], and the DNA sequencing of multiple loci followed by population genetics analyses [26,27,28]. In the presence of recombination, even at a very low rate, Muller’s ratchet is significantly alleviated [78]. Using multiple statistical approaches, we also found strong evidence for mitochondrial recombination. Our evidence, based on whole-genome analyses of natural populations of *A. phalloides*, reinforces the idea that filamentous fungal mitochondrial genomes can recombine.

Recombination involves three possibilities: (1) recombination between different mitochondria inherited from different parents (biparental inheritance), (2) recombination between mitochondrial DNA and nuclear-encoded mitochondrial genes, and (3) recombination between different mitochondrial genotypes found in a single mycelium but derived from somatic mutation [72]. We hypothesize biparental inheritance (1) as the explanation for the observed recombination in *A. phalloides* because the numerous mutations in the 79–82 kbp region (Figure 1 and Figure 3) of assembled mitochondria are not likely to derive from somatic mutations accumulated in a single individual (eliminating the second possibility), and because no putatively homologous sequences of the 79–82 kbp region are found in the nuclear assembly (eliminating the third possibility). 

However, the recombination rate of *A. phalloides* mitochondria is low. We estimated a rate of one recombination every 8–67 mutations. This recombination-to-mutation rate is lower than most estimates for the nuclear genome, for putatively asexual prokaryotes [6], and for the mutation rate of *S. cerevisiae* [79]. However, in a simulation published by Charlesworth et al. (1993), a recombination-to-mutation rate of 2 × 10^−5^ can significantly reduce the rate of Muller’s ratchet, given a high enough effective population size (>100) [78]. Thus, we consider the recombination we observe in the mitochondria of *A. phalloides* as a plausible mechanism for facilitating an escape from Muller’s ratchet.

The low recombination rate appears to explain the low genetic diversity (π and N_e_µ_s_) in the mitochondrial genome compared with the nuclear genome, a phenomenon also reported in other fungal species [27,29,30]. With limited recombination, all mitochondria are at high linkage disequilibrium. When a deleterious mutation emerges, selection against the mutation decreases the effective population size at linked sites, resulting in low genetic diversity [80]. An alternative hypothesis is a lower mitochondrial mutation rate, compared with the nuclear mutation rate. However, this hypothesis seems less likely, based on evidence from studies in other systems, where mitochondrial mutation rate estimates are (mostly) significantly higher than nuclear mutation rate estimates [3,4].

The low recombination rate may also explain the higher π_n_/π_s_ we observe in the mitochondrial genome compared with the nuclear genome: because a low recombination rate can reduce the effective population size, and a reduced effective population size can cause stronger genetic drift, lower negative selection efficacy may follow [7]. However, we cannot rule out the possibility that a high π_n_/π_s_ is simply a result of the strength of selection on the mitochondrial-encoded genes. 

### 4.3. Uniparental Inheritance as an Alternative Strategy to Combat Muller’s Ratchet

The heteroplasmy required for recombination can reduce the variation in fitness among individuals and therefore reduce the efficacy of selection [13,15]. As a result, at mutation–selection–drift equilibrium, mitochondria inherited uniparentally and experiencing strong cellular-level bottlenecks experience efficient selection, resulting in fewer standing deleterious mutations compared with mitochondria inherited biparentally and undergoing recombination [13,15]. We found little evidence for heteroplasmy derived from biparental inheritance. Only one genome, from mushroom 10299, has an estimated heteroplasmy rate higher than 1%, but the estimated heteroplasmy of other mushrooms belonging to the same genetic individual is approximately 0.05%. Moreover, these mushrooms developed from an unusual homokaryotic mycelium, and a homokaryotic mycelium is more unlikely to house genetically different mitochondria, compared with a typical heterokaryotic mycelium. We conclude that the apparent heteroplasmy derives from background contamination and is not evidence for biparental inheritance. A dominant pattern of uniparental inheritance also aligns with most previous research with filamentous basidiomycetes [18,19,20,21]. 

Nonetheless, the evidence for recombination demands the presence of genetically different mitochondria within the same cell, even if heteroplasmy is rare or transient (or both). In *Agrocybe aegerita*, heteroplasmy can only be detected in vegetative hyphae [22]. Following heteroplasmy, however it occurs, subsequent bottlenecks of mitochondria may restore mycelia to homoplasmy and maintain the heterogeneity of fitness among individuals, and the efficacy of selection may still be strong enough to alleviate Muller’s ratchet [13,14]. Of course, it is also possible that heteroplasmy in natural populations of *A. phalloides* mushrooms is more common than we have discovered, perhaps because of the stringency of our analyses, or perhaps we did not capture the full spectrum of potential mitochondrial donors in our test for heteroplasmy. More comprehensive sampling and deeper sequencing will increase the statistical power of future analyses by identifying additional putative parents and may enable the detection of rare mitotypes within a single mushroom. However, additional evidence against routine heteroplasmy is provided by comparisons of mushrooms generated from a single genetic individual; no single mycelium produced mushrooms with different mitotypes, not even the genetic individual represented by 19 mushrooms (and this individual is 10 m across at its widest point). The presence of a single mitotype in each genetic individual is another signature of the uniparental inheritance of mitochondria. 

### 4.4. Mitochondrial Genomes Provide Evidence for a 28-Year-Old Homokaryotic Individual

The mitochondrial dynamics of an unusual homokaryotic individual found in California [39] provide additional clues as to the dynamics of unisexuality in nature. Using a kinship algorithm [81], Wang et al. identified six other (heterokaryotic) genetic individuals as closely related to a homokaryotic individual: these six (heterokaryotic) individuals appear to have one nucleus identical to the nucleus of the homokaryon and are either the offspring of it, or one may be its parent [39]. Five of these offspring/parent house the same mitotype as the homokaryon, and one has a different mitotype. Wang et al. hypothesized that the origin of the homokaryotic individual was a single spore that germinated and began to grow as a monokaryon, and because we have no evidence that death cap mycelia grow as mitochondrial mosaics, the parent of the homokaryon (the mycelium generating the spore) must have the same mitotype as the homokaryon. Therefore, the individual with the different mitotype cannot be the parent of the homokaryon and must be an offspring. Because this offspring was collected in 1993 and the homokaryon was last collected by Wang et al. in 2021, we can confirm that the homokaryon must have been growing as a homokaryon by 1993 and is likely older than 28 years old [39]. Moreover, the fact that five of six matings involving the homokaryon resulted in heterokaryons with the same mitotype as the homokaryon provides tantalizing but incomplete evidence of a transmission advantage for the mitochondria of the homokaryon.

## 5. Conclusions

We explored mitochondrial dynamics in the filamentous basidiomycete *Amanita phalloides* using population genomics data collected from multiple natural populations of the fungus. We discovered that mitochondrial genome sizes are dynamic and can be driven by homing endonuclease activity, as well as large indels of other genetic components. Low levels of recombination likely result in lower nucleotide diversity and a higher π_n_/π_s_ ratio, compared with nuclei. However, the uniparental inheritance of mitochondria may maintain selection efficacy in mitochondria. Our data and interpretations synthesize previous research on fungal mitochondrial inheritance and evolutionary dynamics and extend our understanding of how the mitochondria of filamentous basidiomycetes combat Muller’s ratchet.

## Figures and Tables

**Figure 1 jof-09-00476-f001:**
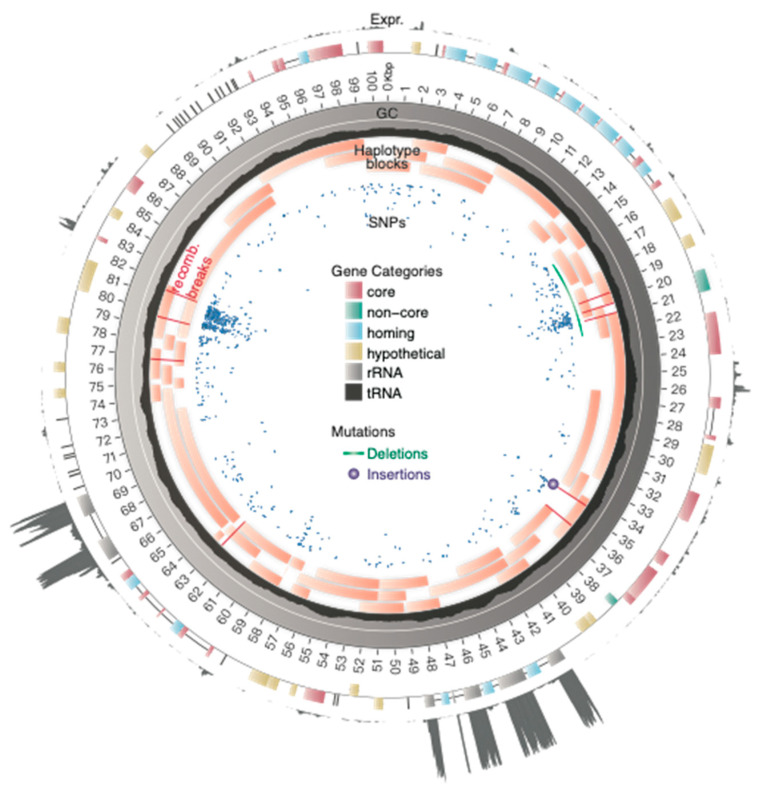
The genome architecture of *Amanita phalloides* mitochondria. From the outermost to the innermost ring: (1) sequencing depth of expression data; (2) annotated genes; lines are introns; (3) physical map; (4) GC content: window size = 1000 bp, step = 10 bp; (5) haplotype blocks and recombination breaks; (6) SNPs and indels.

**Figure 2 jof-09-00476-f002:**
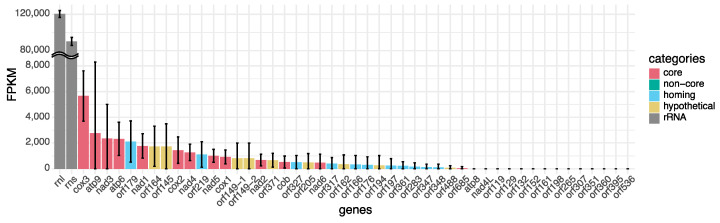
The expression level of each mitochondrial gene (calculated as fragment per kilobase of transcript per million mapped reads).

**Figure 3 jof-09-00476-f003:**
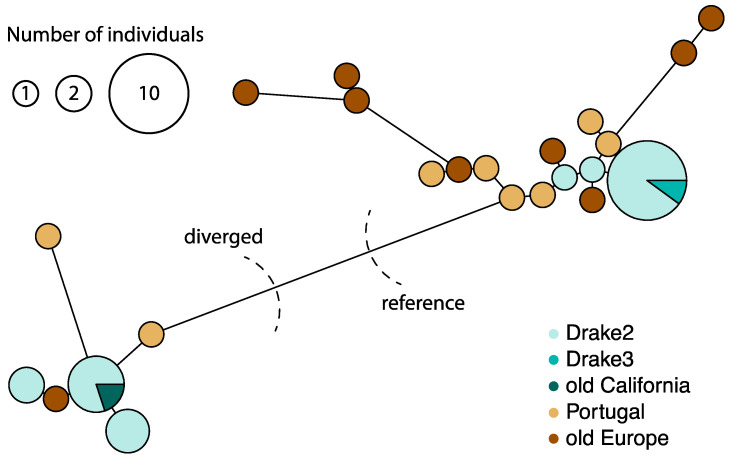
Haplotype network of *Amanita phalloides* mitochondria. Two distinct groups are labeled as either “reference” or “diverged”.

**Figure 4 jof-09-00476-f004:**
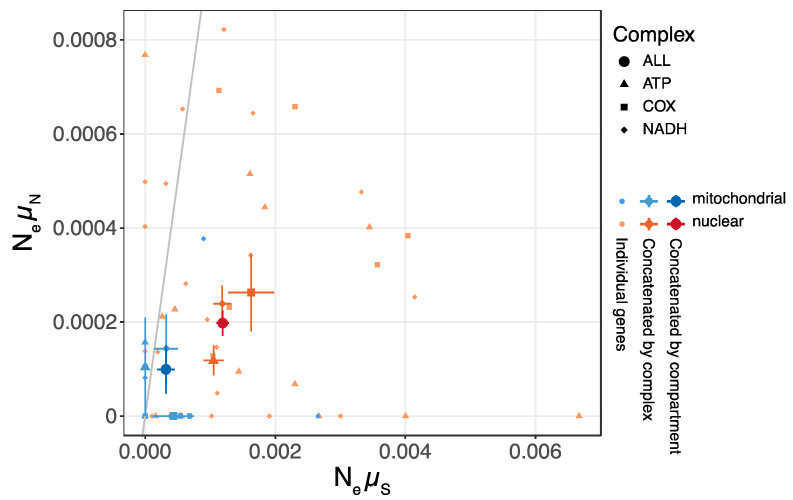
Population synonymous and nonsynonymous mutation rates (N_e_µ_s_ and N_e_µ_n_). Rates calculated without outlier gene *NAD6*. Error bars are standard errors. Diagonal line = π_n_/π_s_ = 1.

**Figure 5 jof-09-00476-f005:**
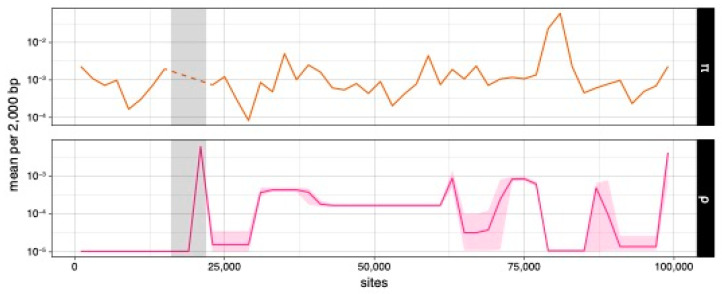
Nucleotide diversity (π) and population recombination parameter (ρ) estimates every 2000 bp. Light-gray areas mark the region with complexed indels. Light-pink ribbon (ρ) marks the 95% credible interval of the estimates. Dashed line indicates estimates not estimated.

**Figure 6 jof-09-00476-f006:**
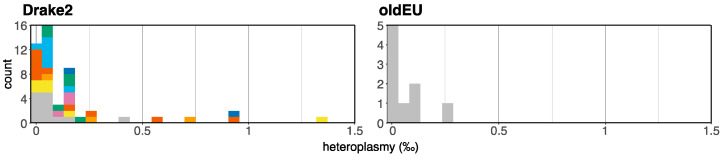
Heteroplasmy estimates for Drake2 mushrooms and a collection of European mushrooms from herbaria (as a control). Except for gray, colors indicate mushrooms belonging to the same genetic individual (mushrooms generated from the same mycelium). Gray: genetic individuals consisting of a single mushroom.

**Table 1 jof-09-00476-t001:** Tests for recombination in the mitochondrial genome.

	Whole Alignment	70~95 kbp
LDr^2^ 10 kbp *	7/10	-
LDr^2^ 20 kbp *	4/5	-
LDD’ 10 kbp *	1/10	-
LDD’ 20 kbp *	3/5	-
Maximum χ^2^ **	0	0
PHI **	**-**	0.86

* LD-distance correlation tests were conducted on 10 or 20 kbp chunks of the mitochondrial genome, and we report the numbers of tests with *p*-values < 0.05 over total numbers of tests. ** Maximum χ^2^ and PHI tests were based on the entire alignment, and so we report *p*-values.

## Data Availability

The datasets generated and/or analyzed during the current study are available in the Open Science Framework repository, [https://osf.io/z7fy8/?view_only=18f56a970a90482ebe999a6602d50564 (last modified on 1 February 2023)].

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
