# Peer review of "Uniparental Inheritance and Recombination as Strategies to Avoid Competition and Combat Muller’s Ratchet among Mitochondria in Natural Populations of the Fungus Amanita phalloides"

_jof, 2023, doi:10.3390/jof9040476_

Round 1

Reviewer 1 Report

As far as I know, this work is great! Exploring mitochondrial evolution in nature is very important, especially for macrofungi. In the future, the authors can create a population from spore monokaryons or protoplast monokaryons by mon-mon hybridization or mon-di hybridization. It would be an alternative method for evaluating the single-parent inheritance and variation of mitochondria by artificial population.

Author Response

As far as I know, this work is great! Exploring mitochondrial evolution in nature is very important, especially for macrofungi. In the future, the authors can create a population from spore monokaryons or protoplast monokaryons by mon-mon hybridization or mon-di hybridization. It would be an alternative method for evaluating the single-parent inheritance and variation of mitochondria by artificial population.

We thank you for the kind words on our manuscript. We would love to test mitochondrial inheritance in vitro. However, culturing Amanita phalloides has proved to be challenging, so we are not able to perform these experiments at the current stage.

Reviewer 2 Report

"Uniparental inheritance and recombination as strategies to avoid competition and combat Muller’s ratchet among mitochondria in natural populations of the fungus Amanita phalloides"

is an important article exploring the balance between competition organelles of different parentage and the mutation load stemming from lack of recombination in fungi. They show that even in fungi, where there are hyphae with free flowing cytoplasm, evolutionary forces on one side prevent the coexistence of organelles that are, however, released from the pressure of mutation load by a very low rate of recombination.

The major comment I would request is to add more information about the calculations behind heteroplasmy estimates.

-explain the equation (2) for H_hat with more detail L296-297

-add a supplemental table to support section 3.5 of the results L477-496 and Figure 6 with the H-hat estimates and additional stats for each mitotype and each sample.

Other comments:

L14 apart from some rare events, in plants and animals the inheritance of mitochondria are uniparental. If the authors want to compare the rates of recombination with those estimated from those studies, that would be very interesting!

L20 Mitochondrial genomes split into two divergent groups -> Mitochondrial genomes clustered into two genetically distinct groups

L202-204 add reason for using average pi over 1kb windows

L211 full stop at the end of sentence

L435-438 Is there an estimate out of the 5.6% for the error rate of biallelic sites the authors could report? 

L444-446 chunks -> windows

L553 drives -> contributes

L563 concept -> idea

L629-630 not a sentence that adds to the paragraph. What dynamic?

Author Response

"Uniparental inheritance and recombination as strategies to avoid competition and combat Muller’s ratchet among mitochondria in natural populations of the fungus Amanita phalloides" is an important article exploring the balance between competition organelles of different parentage and the mutation load stemming from lack of recombination in fungi. They show that even in fungi, where there are hyphae with free flowing cytoplasm, evolutionary forces on one side prevent the coexistence of organelles that are, however, released from the pressure of mutation load by a very low rate of recombination.

We thank you for the kind words on our manuscript. We will respond point-by-point below.

The major comment I would request is to add more information about the calculations behind heteroplasmy estimates.

-explain the equation (2) for H_hat with more detail L296-297

The logic behind equation (2) is that the real detectable heteroplasmy should have a higher read number/K-mer number than those from sequencing errors. We have provided the reasoning at L295–296.

-add a supplemental table to support section 3.5 of the results L477-496 and Figure 6 with the H-hat estimates and additional stats for each mitotype and each sample.

Thank you for the suggestion. We have added the data to supplementary table 1.

Other comments:

L14 apart from some rare events, in plants and animals the inheritance of mitochondria are uniparental. If the authors want to compare the rates of recombination with those estimated from those studies, that would be very interesting!

Thank you for the suggestion. Unfortunately, there are not many estimates for mitochondrial recombination rate available. The only estimate that we are aware of is from Saccharomyces cerevisiae, and we have included a short description at L576.

L20 Mitochondrial genomes split into two divergent groups -> Mitochondrial genomes clustered into two genetically distinct groups

Done.

L202-204 add reason for using average pi over 1kb windows

Thank you for pointing this out. This is a leftover of the previous version of the manuscript and is not included in the current version anymore, so we have removed it.

L211 full stop at the end of sentence

A period is added.

L435-438 Is there an estimate out of the 5.6% for the error rate of biallelic sites the authors could report? 

This is from a single exact count from the dataset, so there is no error rate we can report.

L444-446 chunks -> windows

Done

L553 drives -> contributes

Done

L563 concept -> idea

Done

L629-630 not a sentence that adds to the paragraph. What dynamic?

The sentence refers to the absence of mushrooms with different mitochondrial types within an individual. We have rewritten it to be more clear (Now L627-629).